# The Potential of Thyroid Hormone Therapy in Severe COVID-19: Rationale and Preliminary Evidence

**DOI:** 10.3390/ijerph19138063

**Published:** 2022-06-30

**Authors:** Iordanis Mourouzis, Vassiliki Apostolaki, Athanasios Trikas, Leonidas Kokkinos, Natassa Alexandrou, Maria Avdikou, Myrto Giannoulopoulou, Aimilia Vassi, Ioulia Tseti, Constantinos Pantos

**Affiliations:** 1Department of Pharmacology, Medical School, National and Kapodistrian University of Athens, 75 Mikras Asias Ave., Goudi, 11527 Athens, Greece; imour@med.uoa.gr (I.M.); vapostolaki2013@gmail.com (V.A.); atrikas@otenet.gr (A.T.); kintzios@uni-pharma.gr (I.T.); 2Department of Anesthesiology, ELPIS General Hospital of Athens, 11522 Athens, Greece; marlee@otenet.gr (L.K.); natalexandrou@yahoo.gr (N.A.); avdikou.maria@gmail.com (M.A.); m.yr@virgin.net (M.G.); applevass@yahoo.gr (A.V.)

**Keywords:** thyroid hormone, COVID-19, erythrocyte, sepsis, right ventricle, hypoxia

## Abstract

Tissue hypoxia is one of the main pathophysiologic mechanisms in sepsis and particularly in COVID-19. Microvascular dysfunction, endothelialitis and alterations in red blood cell hemorheology are all implicated in severe COVID-19 hypoxia and multiorgan dysfunction. Tissue hypoxia results in tissue injury and remodeling with re-emergence of fetal programming via hypoxia-inducible factor-1α (HIF-1a)-dependent and -independent pathways. In this context, thyroid hormone (TH), a critical regulator of organ maturation, may be of relevance in preventing fetal-like hypoxia-induced remodeling in COVID-19 sepsis. Acute triiodothyronine (T3) treatment can prevent cardiac remodeling and improve recovery of function in clinical settings of hypoxic injury as acute myocardial infarction and by-pass cardiac surgery. Furthermore, T3 administration prevents tissue hypoxia in experimental sepsis. On the basis of this evidence, the use of T3 treatment was proposed for ICU (Intensive Care Unit) COVID-19 patients (Thy-Support, NCT04348513). The rationale for T3 therapy in severe COVID-19 and preliminary experimental and clinical evidence are discussed in this review.

## 1. Introduction

It is well-recognized that changes in Thyroid Hormone (TH) metabolism occur in response to acute illness, known as non-thyroidal illness syndrome (NTIS). In this syndrome, an initial drop of circulating triiodothyronine (T3) levels is observed, and at later stages a decrease in both circulating L-thyroxine (T4) and T3 levels occurs. Numerous studies have reported that NTIS in acute illness is associated with poor outcomes and high mortality [1]. Similarly, changes in TH metabolism have been observed in COVID-19 patients with an impact on disease progression and outcome [2]. The management of this condition still remains controversial.

Tissue hypoxia due to disturbed microcirculation seems to be a unifying pathophysiologic mechanism in a spectrum of clinical conditions such as myocardial infarction, sepsis and trauma [3]. In this context, TH signaling is shown to be implicated in hypoxia- induced cardiac remodeling [4] and T3 therapy has favorable effects on postischemic cardiac function and infarct healing both in experimental studies and in humans [5,6]. Along this line, T3 has recently been shown to prevent tissue hypoxia in experimental sepsis [7].

On the basis of this evidence, the potential of T3 therapy in severe COVID-19 has been proposed and a clinical trial was initiated [8,9]. In this review, we discuss the rationale for the use of TH as a potential therapy in severe COVID-19 in the view of recent experimental and clinical data.

## 2. Tissue Hypoxia: A Unifying Pathophysiologic Mechanism in Myocardial Infarction, Sepsis and Severe COVID-19

Impairment of microcirculation remains one of the main causes of tissue hypoxia and organ dysfunction in acute critical illness. Despite hemodynamic restoration and oxygen availability, there is a lack of improvement in organ perfusion both in sepsis and ischemia–reperfusion, such as myocardial infarction and by-pass surgery, due to the impaired microcirculation [10,11]. Persistent microcirculatory changes are associated with organ failure and death in patients with septic shock [12]. Similarly, microvascular obstruction after acute myocardial infarction is associated with heart failure and high mortality [13,14]. Along this line, changes in microcirculation were found in patients with severe COVID-19 [15].

Endothelial injury and altered blood cell hemorheology are considered important pathogenetic mechanisms which may lead to impaired microcirculation [16]. Endothelial injury appears to be common in sepsis and myocardial ischemia–reperfusion injury as in myocardial infarction and by-pass surgery [3]. Syndecan-1 and thrombomodulin are the most studied biomarkers of endothelial integrity and function. Syndecan is a heparin sulfate proteoglycan expressed in endothelial cells and the main marker of endothelial glycocalyx degradation [17]. Thrombomodulin is a type I transmembrane glycoprotein that is present on the luminal surfaces of endothelial cells. Plasma levels of syndecan-1 and thrombomodulin are higher in patients with sepsis, and both biomarkers predicted the risk of circulatory failure or death [18]. Similarly, in by-pass surgery, a clinical setting of ischemia–reperfusion injury, acute microcirculatory perfusion changes persisted in the first three postoperative days and were associated with increased heparin sulphate and Syndecan-1 levels [10]. Interestingly, COVID-19 patients with high levels of Syndecan-1 had significantly higher levels of thrombomodulin, interleukin-6 and TNF-α and higher mortality [19]. Furthermore, pulmonary vascular endothelialitis was reported from autopsies of COVID-19 patients [20].

Erythrocyte abnormalities are common in sepsis as well as in ischemia–reperfusion injury and have an impact on microcirculation. Erythrocyte aggregation initially occurs in areas of slow flow and impairs microcirculation. Impaired vascular flow enhances erythrocyte aggregation (a vicious circle), and this in turn causes sludging of blood and thrombosis [16]. Moreover, endothelial nitric oxide synthase (NOS) expression is decreased due to lower shear stress resulting from the axial accumulation of red blood cells (RBCs) [21]. RBC aggregation can disrupt the endothelial glycocalyx as the aggregates rub against and induce endothelial injury [22]. Erythrocyte aggregation occurs in experimental sepsis [23]. The erythrocyte sedimentation rate (ESR), an indirect index of erythrocyte aggregation, is elevated in sepsis. In acute myocardial infarction, patients with a high ESR have a greater risk for cardiovascular death and major adverse cardiovascular events (MACEs) [24]. Erythrocyte aggregation may also occur in COVID-19 [25]. In vitro studies showed a strong contribution of plasma fibrinogen in RBC hyperaggregation. RBC aggregation correlated positively with clot firmness, negatively with clot formation time and positively with the length of hospitalization. Patients receiving oxygen supplementation had higher RBC aggregation and blood viscosity. In addition, patients with pulmonary lesions had higher RBC aggregation and enhanced coagulation [25]. More recently, erythrocyte reactive oxygen species (ROS) production was shown to be increased and NO release to be decreased in COVID-19 vasculopathy [26,27]. Collectively, it appears that erythrocytes may serve an important role in microcirculation and tissue hypoxia in critical illness and particularly in severe COVID-19. Figure 1.

Tissue hypoxia can lead to tissue injury and remodeling, ultimately followed by organ failure. Interestingly, local lung hypoxia determines epithelial fate decisions during alveolar regeneration [28]. In COVID-19, troponin, a biomarker of cardiac injury, was elevated in 30% of patients and associated with increased mortality [29]. Furthermore, elevated troponin was correlated with right ventricular (RV) dysfunction [30]. Tissue hypoxia can induce tissue remodeling by activating fetal-like cellular reprogramming [31]. This response is common after sepsis, myocardial infarction and trauma and involves HIF-1α-dependent and-independent regulated mechanisms [32,33]. Hypoxia-Inducible Factor 1α (HIF-1α) seems to be activated when oxygen partial pressure levels in tissue are below 10 mmHg [32]. HIF-1α plays a key role in regulating metabolic pathways and inflammatory responses [34,35]. Hypoxia-induced changes in organs and immune response have been considered a key feature in COVID-19 pathophysiology [9,36]. Notably, in SARS-CoV-2-infected Caco-2 cells, virus replication, cell cytotoxicity and cytokine production were promoted by a HIF-1α inducer and attenuated by a HIF-1α inhibitor [37].

Although hypoxia-induced apoptosis and remodeling remain the main pathophysiologic mechanisms of serious clinical conditions, effective therapies are not currently available. However, the recognition that hypoxia can induce pathologic remodeling in injured tissues via activation of fetal reprogramming has revived the interest in cell-based therapies and hormones, such as TH, which play a key role in organ maturation during development. In this context, the role of TH and its potential therapeutic use has been excessively studied in myocardial ischemia [38]. At this point, it should be noted that low TH can result in cardiac microvascular impairment and rarefaction [33,39]. Furthermore, in an experimental model of sepsis, down-regulation of TH signaling was associated with a decrease in the total number of mitochondria, an increase in the percentage of injured mitochondria and downregulation of respiratory chain complex 2 and 3 mRNA expression, resulting in reduced oxidative phosphorylation [40].

## 3. TH and Tissue Hypoxia

### 3.1. Differential Effects of TH on Healthy and Injured Tissue

The use of TH therapy in clinical practice has been limited due to the long-standing belief that TH can aggravate myocardial ischemia by increasing the metabolic rate and oxygen demand. However, over the past years, landmark experimental studies have challenged this long-standing belief, showing that the effects of TH are different in normal and diseased states (recently reviewed in [33]). In an isolated rat heart model, we previously found that acute, high-dose T3 administration at reperfusion after zero-flow global ischemia improves recovery of function and reduces myocardial injury, such as apoptosis and necrosis [41,42]. In contrast, the same dose of T3 in normal isolated perfused hearts did not have any effect on myocardial function. In accordance with this, a recent study has also shown that high-dose T3 administration at reperfusion can reduce infarct size and improve mitochondrial function [43]. Furthermore, high-dose T3 increased contractile recovery in human right atrial trabeculae subjected to hypoxia and reoxygenation [44].

Similar observations have been reported for other organs, such as the kidney and liver, in different experimental settings. TH treatment appears to induce protection against ischemic injury in the kidney [45,46]. T3 treatment 24 h prior to renal ischemia–reperfusion resulted in a marked decrease in proteinuria [45]. In addition, T3 treatment prevented acute tubular necrosis after renal ischemia–reperfusion injury [47]. T3 treatment in experimental sepsis prevented cardiac and liver tissue hypoxia (pO2 less than 10 mmHg, a threshold below which HIF-1α-regulated mechanisms are activated) [7]. On the contrary, high-dose T3 administration in normal animals for 10 days or 7 weeks resulted in increased oxygen consumption, intrarenal tissue hypoxia and proteinuria [48,49]. These data clearly indicate that TH effects are different in stressed tissue than in normal.

### 3.2. Potential Underlying Mechanisms of TH Effects on Tissue Hypoxia

TH regulates the microvascular function both directly and indirectly. T3, acting through thyroid hormone receptor α (TRα), induces endothelium-dependent vasodilation via endothelial nitric oxide synthase [50]. The increased metabolic rate induced by TH may also lead to vasodilation indirectly. Furthermore, TH can induce physiologic angiogenesis, increasing new small vessels with normal permeability and function via direct regulation of angiogenic factors such as Vascular Endothelial Growth Factor-A, Fibroblast Growth Factor-2, angiopoietin-2 and Platelet-derived Growth Factor [33]. TH may also increase tolerance to hypoxic injury via induction of adaptive molecules such as the Heat Shock Proteins [51]. Additional evidence suggests that TH interacts with HIF-α and the hypoxia response pathway. This action may have profound effects on the HIF-induced organ remodeling and dysfunction [33,52].

### 3.3. TH as Potential Therapy for Tissue Hypoxia

The therapeutic effect of TH on cardiac tissue hypoxia has been tested in experimental models and recently in clinical trials. TH pre-treatment can protect against myocardial ischemia–reperfusion injury in a pattern similar to ischemic pre-conditioning [53]. High-dose T3 and not T4 administration at reperfusion improved cardiac function and limited apoptosis, particularly in the mid-layer of the myocardium, an area where microvascular circulation prevails [42]. This effect required the TRα1 receptor [41] and was mediated by up-regulation of pro-survival Akt and suppression of p38 MAPK [42].

Post-ischemic cardiac remodeling appears to be associated with distinct temporal changes in TRs [54], and an interaction between (TRs) and adrenergic or inflammatory signaling was shown to occur in cell-based experimental models [51]. Accordingly, TH administration in a dose-dependent manner prevented cardiac remodeling in experimental models of myocardial infarction [55]. This experimental evidence was recently translated into a pilot randomized clinical trial (Thy-Repair) which investigated the effects of early high-dose L-triiodothyronine (LT3) in patients with anterior MI undergoing primary angioplasty [6]. In accordance with the experimental evidence, this study showed that LT3 can prevent dilatation of the left ventricle (LV), electrical remodeling and can also improve infarct healing, without any major adverse effects. A favorable effect on microvascular obstruction (MVO) was also observed [6]. In addition, early T3 administration in by-pass surgery was shown to improve cardiac function and limit troponin release, too [56].

Along this line, T3 therapy has favorable effects on cardiac dysfunction due to excessive inflammation driven by TNFα overexpression [57]. TNFα overexpression resulted in down-regulation of TRβ1 as previously found in cell-based experimental models [58]. Interestingly, in this model, a gender-specific reduction in T3 levels could cause the worst cardiac phenotype observed in female mice, and T3 administration improved cardiac function and calcium handling via controlled Akt activation [57].

## 4. The Role of TH in COVID-19

There is accumulating evidence showing that alterations in TH metabolism frequently occur in COVID-19 patients. In spite of the relatively high heterogeneity between studies, low TH levels on admission were associated with COVID-19 disease severity and mortality. Furthermore, the probability of finding low Free T3 (FT3) and low Thyroid-Stimulating Hormone (TSH) levels was related to disease severity in patients without known thyroid disease. It should be noted that blood samples, in most of the studies reported in the literature, were collected on patients’ admission or during the first three days of hospitalization either in the ward or in ICU.

A retrospective study of 46 patients evaluated the thyroid function in two different groups of consecutive patients affected by COVID-19, one with pneumonia and the other with acute respiratory syndrome (ARDS) requiring ICU admission in comparison with euthyroid patients. COVID-19 patients showed a statistically significant reduction in FT3 and TSH levels measured one day after admission, and as far as ICU patients were concerned, a further statistically significant reduction in FT3 and TSH was found. These findings probably indicate a negative association between FT3 and TSH levels with disease severity at initial presentation [59].

Another cohort study by Lui et al. [60], after a follow-up of 191 patients with COVID-19, reported that around 15% of patients with mild-to-moderate disease had abnormal thyroid function. A decreasing trend of FT3 with increasing COVID-19 severity was found, and patients with low FT3 had more adverse COVID-19-related outcomes.

In a cohort observational study of 196 patients, an NTIS pattern (low T3 with low or normal TSH) was observed in 60% of ICU and 36% of ward patients. TSH, FT4 and T3 levels were measured within 24 h of admission and related to the severity of the disease, while the incidence of NTIS between SARS-CoV-2-positive and -negative patients was not different [61]. Furthermore, FT4 levels were similar between ICU and ward patients.

In a retrospective study including 236 patients, low FT3 on admission independently correlated with the severity of COVID-19 [62]. Similar conclusions were reached by the studies of Gao et al. [63] and Schwartz et al. [64]. FT3 levels within 3 days of either ICU or non-ICU patients appeared to predict disease severity in the early presentation of COVID-19 [64]. More importantly, mortality rates were also higher in the subgroup of patients with low FT3.

In a prospective study of 250 hospitalized COVID-19 patients, both FT3 and FT4 levels on admission were found to be lower in critically ill ICU patients compared to ward patients [65]. Chen et al. also reported lower T3 and TSH but not T4 levels (measured within 3 days after admission) with increasing disease severity in COVID-19 patients [66].

In the study by Campi et al., almost half of the 144 COVID-19 patients had normal TH both on admission and during the 3–7-day follow-up [67]. In 39% of the patients, low TSH levels were found either on admission or during hospitalization, along with low FT3 in half of the cases. TSH and FT3 levels were invariably restored at the time of discharge in survivors, whereas they were persistently low in most deceased cases. Interestingly, only FT3 levels were predictors of mortality. Furthermore, Gong’s retrospective study recruited 150 patients with COVID-19 and divided those patients into two groups of low and normal TSH. Critical illness and mortality rates were significantly higher in the low-TSH than in the normal-TSH group, but it was not significantly different between the low-FT4 and the normal-high FT4 group [68].

Along this line, a retrospective observational study of 127 COVID-19 patients showed that serum FT3 levels on admission were lower in non-survivors compared to survivors among moderate-to-critical patients, and the low FT3 state on admission was associated with an increased risk of all-cause in-hospital mortality in these patients [69]. Furthermore, in a retrospective study of 78 critically ill COVID-19 patients, the levels of FT3 on ICU admission were lower in non-survivors compared to survivors, and patients who survived had higher levels of FT3, FT4 and TSH than non-survivors by day 5 [70]. However, in a recent meta-analysis including 3609 COVID-19 patients, the relation between FT3 or TSH levels and survival could not be reliably assessed due to high heterogeneity between studies. This analysis further showed that survivors had higher FT4 levels than non-survivors [71].

The potential mechanisms implicated in TH abnormalities in COVID-19 are not fully understood. It is likely that changes in the Hypothalamic–Pituitary–Thyroid (HPT) axis and iodothyronine deiodinase activity due to abnormal systemic inflammatory response may occur. Furthermore, SARS-CoV-2, via its direct action, can result in destruction of the thyroid gland [71]. Thyroid follicular cells have been shown to express ACE2 (the target of the spike protein), while typical histopathological features of thyroid injury have been found in patients who died of COVID-19 [72,73].

Concomitant medications used in COVID-19 patients may also affect thyroid function. Systemic corticosteroids, mainly dexamethasone, are recommended for patients with severe and critical COVID-19 based on the guidelines of the World Health Organization (WHO) to control the exacerbated inflammatory response [2], though glucocorticoids are known to affect serum TSH levels in humans. A physiological dose of hydrocortisone appears to play a role in the daily variation in serum TSH levels [74]. Acute inhibition of TSH secretion has also been reported after administration of pharmacological doses of glucocorticoids [75].

## 5. TH Therapy in Severe COVID-19: Preliminary Clinical Evidence

On the basis of the above evidence, a pilot randomized clinical trial (RCT) was designed to test the effects of early, high-dose LT3 in COVID-19 patients admitted to the ICU (ThySupport study). The detailed protocol of this study has been previously described [8]. Co-morbidities were not excluded, and patients with severe systematic diseases (e.g., cancer) with a life expectancy of less than 6 months were not included.

During patient screening, 18 critically ill COVID-19 patients were not included in the study according to trial protocol requirements (e.g., lack of informed consent, sympathomimetic use before initiation of investigational drug, etc.). All these patients were intubated and admitted to the ICU, a complete medical history was recorded, and laboratory tests including total T3, total T4 and TSH were performed during admission to the ICU, as depicted in Table 1. Only 44.4% (8/18) of these patients survived. In this group of patients, total T3 levels on admission were very low in both survivors and non-survivors, while total T4 levels were significantly higher in patients who survived. The decrease in T3 levels came along with a low or normal TSH, suggesting that the HPT axis was probably dysregulated. A total of 88.9% (16/18) of these patients received dexamethasone before admission in the ICU, indicating that corticosteroid treatment for excessive inflammation did not improve total T3 levels and may also have contributed to TH suppression. Furthermore, higher total T4 levels were related to favorable outcomes in this patient population, a finding which is in accordance with previous reports [71].

The Thy-Support study is the first study which was designed to test the potential effects of LT3 administration after intubation in patients with severe COVID-19. Despite the limited number of patients included in the first interim analysis, important novel findings were observed, which warrant further investigation in future clinical trials. In this clinical setting, LT3 was used in a similar therapeutic regimen as in patients with myocardial infarction [6], and baseline patient details have been previously reported [76].

Preliminary data from the Thy-Support study showed that LT3 treatment of ICU COVID-19 patients resulted in an acute drop in the ESR within 48 h of its administration along with a trend of a decrease in troponin levels [76]. Furthermore, a strong correlation between circulating T3 levels and the ESR was found. This novel observation may be of physiological and therapeutic relevance. The ESR is considered an indirect index of erythrocyte aggregation and red blood cell rheology, and thus, these data may indicate a potential effect of LT3 on red cell aggregation, which merits further investigation. Erythrocyte aggregation is an important pathophysiologic mechanism in the vasculopathy of COVID-19, and effective treatments targeting red blood aggregation are not currently available [26].

Here, from the same patients included in Thy-Support, we provide additional findings probably indicating a novel effect of LT3 treatment on functional indices of the right ventricle (RV) in ICU COVID-19 patients. RV dysfunction is detected in nearly 39% of COVID-19 patients and is associated with a worse prognosis [77]. RV dysfunction appears to be prominent in patients with elevated troponin, indicating poor tolerance of overloaded RV to tissue hypoxia [30]. In the Thy-Support study, echocardiography was used to assess functional indices in ICU patients with COVID-19 [8]. The tricuspid annular plane systolic excursion (TAPSE) to systolic pulmonary artery pressure (PASP) ratio was used as an index of coupling of RV to pulmonary artery circulation. This index seems to be an independent predictor of survival in COVID-19 ARDS [78]. Furthermore, in a cohort of 133 COVID-19 patients who underwent full echocardiographic analysis and follow-up, the best cut-off for predicting in-hospital mortality was the TAPSE/PASP ratio < 0.57 mm/mmHg and associated with more than a 4-times increased risk of death [79]. In the Thy-Support study, the TAPSE/PASP ratio was found to be above the cut-off of 0.57 mm/mmHg in patients treated with LT3. Figure 2A. In fact, the TAPSE/PASP ratio 48 h after treatment was 0.85 (0.18) in the LT3 group vs. 0.55 (0.15) in the placebo group, *p* = 0.3. Furthermore, a strong inverse correlation was found between the ESR and TAPSE/PASP ratio at 48 h (r = −0.8, *p* = 0.06); see Figure 2B. Left ventricular function was not abnormal, and it was comparable in the two groups (LVEF% was 60% (3.0) for LT3-treated patients and 65% (5.0) for the placebo, *p* = 0.31).

Collectively, these data probably indicate that LT3 in severe COVID-19 can improve RV–pulmonary artery circulation coupling, and this, at least in a part, may be mediated via changes in the erythrocyte aggregation and hemorheology. Historically, the link between hyperviscosity and right ventricular dysfunction was recognized by Sir William Osler, who treated the RV dilatation with phlebotomy [16].

The Thy-Support study has been prematurely discontinued due to continuously changing therapeutic protocols for management of COVID-19 patients. These modifications in therapeutic protocols negatively affected patients’ recruitment. The Thy-Support study was designed to investigate the effects of T3 administered in COVID-19 patients with early mechanical respiratory support before the development of multiorgan failure and hemodynamic deterioration requiring high-dose vasoactive agents. However, after the use of novel therapies (e.g., anti-virals, corticosteroids) patients’ mechanical respiratory support was delayed [80] and a great number of patients admitted to the ICU were in severe clinical condition as shown in Table 1. Thus, the ThySupport study had to be redesigned. Based on this preliminary clinical evidence and our experimental data, a new phase IIa study is under design to investigate the effects of T3 therapy early in sepsis due to respiratory infections including COVID-19.

## 6. Conclusions

Recent research has revived the interest in the use of TH for combating hypoxia- induced injury and tissue remodeling. TH, once considered to be detrimental in hypoxia, is now realized to have reparative actions after hypoxic injury via evolutionary conserved mechanisms. In this context, experimental and clinical evidence signals for further investigation of the potential use of TH in sepsis and particularly COVID-19. Favorable effects of acute LT3 treatment have recently been reported in patients with myocardial infarction (Thy-Repair) without major adverse effects [6].

## 7. Patents

The following patents are relevant to the work in this manuscript.

PCT/EP2019/087056. L-triiodothyronine (T3) for use in limiting microvascular obstruction.

PCT/4972/2021. A pharmaceutical composition comprising L-triiodothyronine (T3) for use in the treatment of tissue hypoxia and sepsis.

## Figures and Tables

**Figure 1 ijerph-19-08063-f001:**
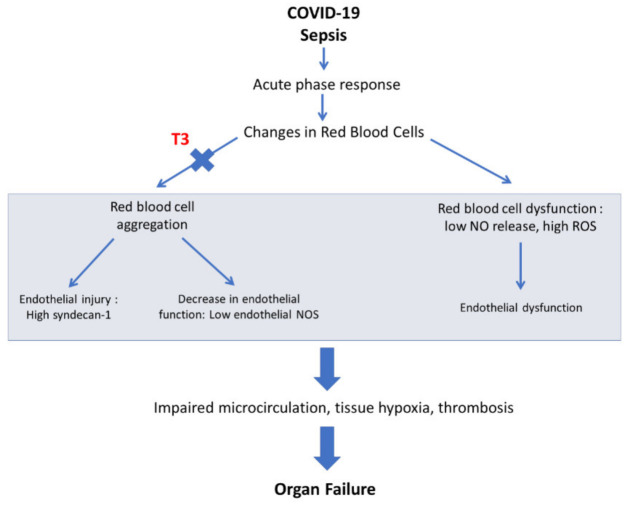
Schematic of the potential role of erythrocyte in COVID-19 vasculopathy and the potential effect of T3 in this response. T3: triiodothyronine, ROS: reactive oxygen species, NO: nitric oxide, NOS: NO synthase.

**Figure 2 ijerph-19-08063-f002:**
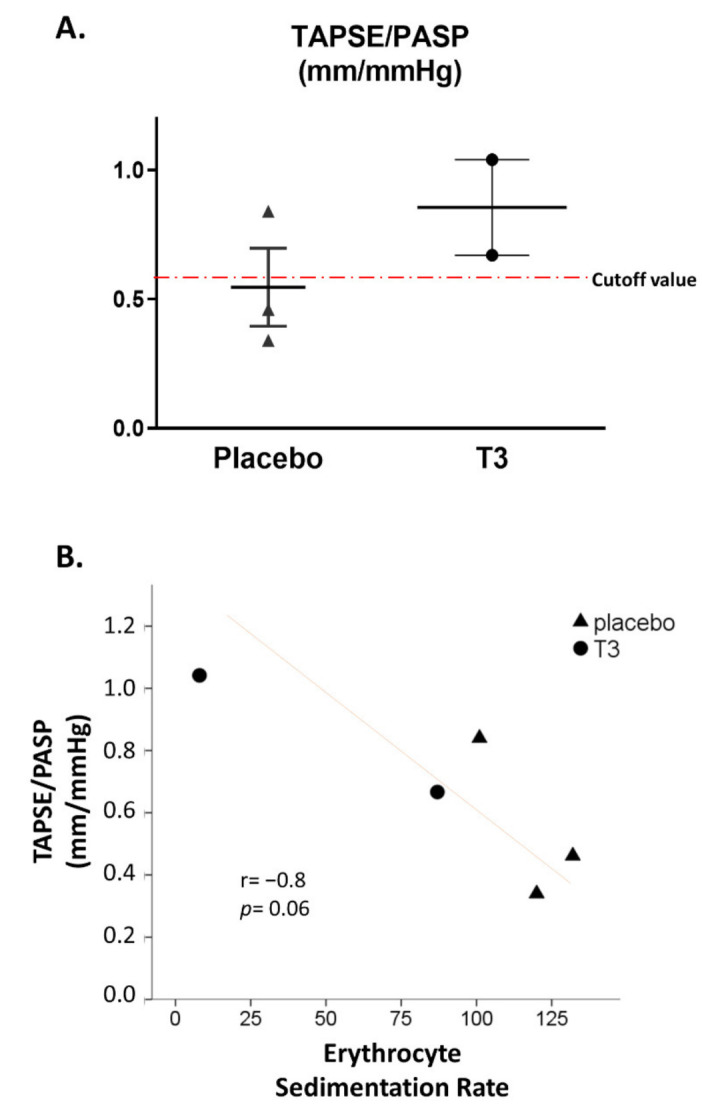
(**A**) Tricuspid annular plane systolic excursion (TAPSE) to systolic pulmonary artery pressure (PASP) ratio for placebo and T3-treated patients at 48 h in COVID-19 patients recruited in Thy-Support trial is shown. Please note that T3-treated patients are above the cut-off value of 0.57, which is reported to determine prognosis in these patients. (**B**) A strong inverse correlation seems to exist between erythrocyte sedimentation rate with TAPSE/PASP ratio at 48 h in COVID-19 patients recruited in Thy-Support trial. T3: triiodothyronine.

**Table 1 ijerph-19-08063-t001:** Thyroid hormone levels and other parameters of critically ill COVID-19 patients at the first day of admission in ICU. Data were collected during screening for the pilot RCT Thy-Support.

	Survivors (*n* = 8)	Non-Survivors (*n* = 10)	Sign. (*p*)
Age (years)	64.4 (8.7)	68.1 (12.1)	0.47
Gender (Male/Female)	4/4	6/4	0.67
Co-morbidities	75% (6/8)	100% (10/10)	0.18
Dexamethasone treatment	100% (8/8)	80% (8/10)	0.47
P/F ratio *	152 (71)	114 (43)	0.17
Lactate (mmol/L)	0.95 (0.33)	2.35 (3.1)	0.23
D-Dimers (ng/mL)	1563 (1403)	2443 (2869)	0.41
Fibrinogen (mg/dL)	315 (329)	471 (28)	0.4
WBC (cells × 103) *	11.85 (5.57)	13.84 (6.77)	0.51
Troponin (pg/mL)	188 (451)	445 (704)	0.41
Total T3 (ng/mL) *	0.46 (0.09)	0.44 (0.12)	0.73
Total T4 (μg/dL) *	7.53 (1.0)	5.27 (2.1)	0.027
TSH (μIU/mL) *	1.0 (1.33)	0.28 (0.32)	0.16

* P/F ratio: PaO_2_/FiO_2_, WBC: White Blood Cells, T3: Triiodothyronine, T4: L-thyroxine, TSH: Thyroid Stimulating Hormone.

## Data Availability

The datasets used and/or analysed during the current study are available from the corresponding author on reasonable request.

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
