# Peer review of "The Potential of Thyroid Hormone Therapy in Severe COVID-19: Rationale and Preliminary Evidence"

_ijerph, 2022, doi:10.3390/ijerph19138063_

Round 1
Reviewer 1 Report
The interesting manuscript “The potential of thyroid hormone therapy in severe COVID-19: Rationale and preliminary evidence” describes that non-thyroidal illness syndrome often occurs in COVID-19 patients, the mechanism of hypoxia and how thyroid hormones could be useful in the treatment of COVID-19. However, some points should be discussed more deeply, for example: the mechanism by which thyroid hormones improve hypoxia should be detailed. Thyroid hormones (TH) are known to increase basal metabolic rate and thus indirectly lead to vasodilation. However, in some organs/tissues the treatment with thyroid hormones leads to hypoxia, such as in kidney (Friederich-Persson et al., 2013. Adv Exp Med Biol.;789:9-14. doi: 10.1007/978-1-4614-7411-1_2; Sivertsson et al., 2022. PLoS One;17(3):e0264524. doi: 10.1371/journal.pone.0264524). Thus, the difference between the mechanism involved in the beneficial effect of TH on hypoxia in heart versus the detrimental effect on hypoxia in kidney should be detailed. In addition, a discussion about the comorbities in which the treatment with TH in COVID-19 should not be recommended should be included. The authors should better explain the changes in inclusion/exclusion criteria that led to the discontinuation of the Thy-Support study, why, and whether there is any additional initiative to study the possible benefit of TH in the treatment of COVID-19. The manuscript is well-written and within the scope of Int. J. Environ. Res. Public Health.
Author Response
We thank the reviewers for the constructive comments that help us improve our manuscript.
Reviewer 1
- We added a paragraph about the mechanisms by which thyroid hormones improve hypoxia. Lines 150-159
- We added a paragraph about the differential effects of TH on healthy and injured tissue and we added and discussed the references suggested by the reviewer. Lines 125-147
- We clarified that in ThySupport study, co-morbidities were not excluded. Lines 267-268
- We added more detailed information about the reasons that led to the discontinuation of the Thy-Support study and also about the new phase II trial that is under design. Lines 336-348
Reviewer 2 Report
The present paper is well documented and well written. Although, in line 14 the term “recapitulation” might not properly used, please replace it with another more suitable term and in line 76 “a viscous circle” refers probably to vicious circle (if so, please correct the phrase).
Author Response
We thank the reviewers for the constructive comments that help us improve our manuscript.
Reviewer 2
the term “recapitulation” has been replaced with the term ´´re-emergence´´,
The phrase “viscous circle” has been corrected to ´´vicious circle´´